# What effect have NHS commissioners' policies for body mass index had on access to knee replacement surgery in England?: An interrupted time series analysis from the National Joint Registry

**Joanna McLaughlin** [1]*, **Ruth Kipping**[2], **Amanda Owen-Smith**[2], **Hugh McLeod**[2,3], **Samuel Hawley**[1], **J Mark Wilkinson**[4], **Andrew Judge**[1,5,6]

1 Musculoskeletal Research Unit, Translational Health Sciences, Bristol Medical School, University of Bristol, Learning and Research Building, Level 1, Southmead Hospital, Bristol, United Kingdom, 2 Population Health Sciences, Bristol Medical School, University of Bristol, Bristol, United Kingdom, 3 National Institute for Health Research Applied Research Collaboration West (NIHR ARC West), University Hospitals Bristol NHS Foundation Trust, Bristol, United Kingdom, 4 Department of Oncology and Metabolism, The Mellanby Centre for Musculoskeletal Research, University of Sheffield, Metabolic Bone Unit, Sorby Wing, Northern General Hospital, Sheffield, United Kingdom, 5 National Institute for Health Research Bristol Biomedical Research Centre, University Hospitals Bristol and Weston NHS Foundation Trust and University of Bristol, Bristol, United Kingdom, 6 Nuffield Department of Orthopaedics, Rheumatology and Musculoskeletal Sciences (NDORMS), University of Oxford, Oxford, Kingdom

* joanna.mclaughlin@bristol.ac.uk

## Abstract

### Objective

To assess the impact of local commissioners' policies for body mass index on access to knee replacement surgery in England.

### Methods

A Natural Experimental Study using interrupted time series and difference-in-differences analysis. We used National Joint Registry for England data linked to the 2015 Index of Multiple Deprivation for 481,555 patients who had primary knee replacement surgery in England between January 2009 and December 2019. Clinical Commissioning Group policies introduced before June 2018 to alter access to knee replacement for patients who were overweight or obese were considered the intervention. The main outcome measures were rate per 100,000 of primary knee replacement surgery and patient demographics (body mass index, Index of Multiple Deprivation, independently-funded surgery) over time.

### Results

Rates of surgery had a sustained fall after the introduction of a policy (trend change of -0.98 operations per 100,000 population aged 40+, 95% confidence interval -1.22 to -0.74, P<0.001), whereas rates increased in localities with no policy introduction. At three years

**Data Availability Statement:** The dataset regarding the policy position and start for the

clinical commissioning groups has been made available in the supplementary file already provided. Upload of the dataset obtained from the National Joint Registry regarding individual surgeries is not possible due to restrictions in access to the registry's data set by the National Joint Registry Research Sub-committee. Electronic health records are, by definition, considered sensitive data in the UK by the Data Protection Act and cannot be shared via public deposition because of information governance restriction in place to protect patient confidentiality. Access to data is available once approval has been obtained through the individual constituent entities controlling access to the data. Access to data is available from the National Joint Registry for England and Wales, Northern Ireland and the Isle of Man, but restrictions apply to the availability of these data, which were used under license for the current study, and so are not publicly available. Data access applications can be made to the National Joint Registry Research Sub-committee. http://www.njrcentre.org.uk/njrcentre/Research/ Researchrequests/tabid/305/Default.aspx contains information on research data access request to the National Joint Registry.

**Funding:** This study is funded by the National Institute for Health Research (NIHR https://www. nihr.ac.uk/) – JM holds an NIHR Doctoral Research Fellowship (NIHR 301469). AJ was supported by the NIHR Biomedical Research Centre at University Hospitals Bristol and Weston NHS Foundation Trust and the University of Bristol. HM was supported by the NIHR ARC West at University Hospitals Bristol and Weston NHS Foundation Trust. The views expressed in this publication are those of the authors and not necessarily those of the NHS, the National Institute for Health Research or the Department of Health and Social Care. The funders had no role in study design, data collection and analysis, decision to publish, or preparation of the manuscript.

**Competing interests:** All authors have completed the Unified Competing Interest form at www.icmje. org/coi_disclosure.pdf and declare: AJ has received consultancy fees from Freshfields Bruckhaus Deringer and has held advisory board positions (which involved receipt of fees) from Anthera Pharmaceuticals, INC, outside the submitted work; no other relationships or activities that could appear to have influenced the submitted work. This does not alter our adherence to PLOS ONE policies on sharing data and materials.

after introduction, there were 10.5 per 100,000 population fewer operations per quarter aged 40+ compared to the counterfactual, representing a fall of 14.1% from the rate expected had there been no change in trend. There was no dose response effect with policy severity. Rates of surgery fell in all patient groups, including non-obese patients following policy introduction. The proportion of independently-funded operations increased after policy introduction, as did the measure of socioeconomic deprivation of patients.

## Conclusions

Body mass index policy introduction was associated with decreases in the rates of knee replacement surgery across localities that introduced policies. This affected all patient groups, not just obese patients at whom the policies were targeted. Changes in patient demographics seen after policy introduction suggest these policies may increase health inequalities and further qualitative research is needed to understand their implementation and impact.

## Introduction

Knee replacement is a common elective surgical procedure that is highly effective at reducing pain and improving functional outcome in patients with end stage knee osteoarthritis [1,2]. The global prevalence of knee osteoarthritis in individuals aged 40+ is 22.9% [3]. One in 10 people in the UK can expect to receive a knee replacement at some point in their lifetime [4] and approximately 120,000 procedures are performed each year in the UK [5]. Demand is increasing with an ageing population and rising levels of obesity [6]; even before the delays in access to surgery arising from the COVID-19 pandemic, more than half a million people were on the waiting list for elective trauma and orthopaedics in England and Wales [7]. Pathways to surgery across the National Health Service (NHS) are increasingly incorporating 'health optimisation' interventions to encourage eligible patients to lose excess weight [8], most commonly for hip and knee elective surgery pathways. The intended outcomes include a reduction in surgical procedures, improved safety, outcomes and recovery from surgery and taking the wider public health opportunity offered by the 'teachable moment' of surgery to trigger lasting lifestyle change [9]. Health optimisation presents an interplay between rationing for resource preservation and health improvement [10–12]. The Royal College of Surgeons states that commissioning policies should be based on clinical need and not factors such as a patient's weight [13], yet around 70% of England's NHS clinical commissioning groups (CCGs) currently restrict access to joint replacement based on body mass index (BMI) [14].

Policies determining health optimisation practices vary across CCGs in England. Policies range in severity from a recommendation that overweight patients are offered advice on weight management, to mandated extra waiting periods to engage with weight management, through to the most stringent with BMI thresholds for surgical referral [14,15]. Despite the longstanding use of commissioners' policies regarding BMI-based restrictions for knee surgery, few evaluations of their impact have been published and the evidence base for their effectiveness remains unclear [8,16,17].

There is an urgent need to provide UK and international decision-makers with high-quality evidence on the impact of health optimisation interventions, whether they increase inequalities in access to surgery, and whether there are wider public health benefits to be gained by

reshaping or extending their use. Our aim was to understand the impact of policy introduction on inequalities and patient access to elective knee replacement surgery in England. This was achieved using data from the National Joint Registry; we used a natural experimental study design with interrupted time series analyses to model the impact introduction of these polices has had on trends in rates of elective knee replacement surgery. We examined the difference in outcomes between CCGs with and without BMI policies. Our *a priori* hypothesis [18] was that policy introduction would be associated with a reduction (level change) in rate of surgery, without a change in the trend.

## Methods

### Study design

We used a quasi-experimental natural experiment study design [19,20]. We evaluated the impact of the introduction of CCG health optimisation polices on trends before and after implementation of the intervention. The timing of introduction of health optimisation policies varied by CCG.

### Data source

We used data from the National Joint Registry for England, Wales, Northern Ireland and the Isle of Man (NJR). The NJR contains data on all publicly and privately funded knee replacement operations, and includes 2 million patients since 2003, covering 96% of primary knee replacements [5]. It is mandatory for surgeons and their hospital to register all knee replacement activity in the NJR, whether the procedures are funded by the NHS or independently. The NJR contains anonymised patient data on age, gender and date of procedure. Information on the patient's residential area, as defined by the 2011 census Lower Layer Super Output Areas (LSOA) are also available. LSOAs are defined as geographical areas of a similar population size, with an average of 1,500 residents [21]. We used the dataset prepared for the NJR's 2019 annual report [22] which therefore did not require further cleaning or coding. We used data provided by the Office for National Statistics (ONS) to identify the LSOAs nested in each CCG locality [23]. As a measure of socioeconomic deprivation, we used the index of multiple deprivation (IMD) score; a relative measure of deprivation based on LSOAs. We used the IMD rank for a patient's LSOA and categorised patients into quintiles based upon the national ranking of local areas, with quintile 1 being the most deprived group and quintile 5 being the least deprived group. Information on relevant CCG policy content, introduction and cessation dates was gathered in July 2021 through collection of policy documentation from CCG websites supplemented with Freedom of Information requests to each CCG [14].

### Ethics approval

Before Personal Data and Sensitive Personal Data is recorded for the NJR, express written patient consent is provided. The NJR records patient consent as either 'Yes', 'No', or 'Not Recorded'. With support under Section 251 of the NHS Act 2006, the Ethics and Confidentiality Committee (ECC), (now the Health Research Authority Confidentiality Advisory Group) allows the NJR to collect patient data where consent is indicated as 'Not Recorded'.

### Participants and inclusion criteria

The study sample consisted of 605,221 patients who had a primary knee replacement, total or uni-compartmental, between January 2009 and December 2019 in England and recorded in

the NJR. Inclusion criteria were patients age 40+ years with osteoarthritis as a primary reason for surgery.

## Outcome measure

The primary outcome was the rate of provision of primary knee replacement for each CCG. For each annual quarter in each CCG, rates (expressed as per 100,000 persons) of surgery were determined by aggregating the number of eligible primary knee replacement procedures in the CCG locality (numerator) and using the aggregated ONS count of the population aged 40 + years living in each of these CCG localities in 2019 as the denominator [24].

## Intervention

The intervention is the date the CCG introduced a health optimisation policy on access to knee replacement surgery. We considered ≥18 months of data post-policy introduction as sufficient to give time for policy implementation and possible influence of existing waiting lists. CCGs were excluded where their policy start date was unknown, policies were stopped and restarted, or where insufficient post-policy introduction data were available. Details of the policy for each CCG included in the analyses are provided in S1 Table.

## Control

Each CCG that introduced a policy, acted as its own control, through comparison of trends in rates of surgery in the time period before policy introduction and the time period after it was introduced. To account for potential external influencing factors, data from CCGs with no policy introduction over the time period of interest were included to control for secular changes in outcomes, using a difference-in-differences controlled interrupted time series study design [18]. This approach provides a test of the differential effects of the intervention timepoint between the groups.

## Effect modification variables

Analyses were stratified according to: BMI thresholds; IMD deprivation decile, and whether patients received public or privately funded surgery (NHS or independent). For BMI calculations, only the individual records with a BMI record in the range 12 to 60 kg/m$^2$ were retained. To explore heterogeneity according to type of CCG policy, policies were categorised as 1 (mild–patients receive advice only), 2 (moderate–patients are subject to additional waiting time before surgery) or 3 (strict–patients must be below a BMI threshold to be eligible for surgery).

## Statistical analyses

**Before-and-after analysis.** We used interrupted time series analysis to examine the impact of policy introduction by calculating trends in the quarterly rates of knee replacement surgery for each CCG group. Segmented linear regression models were used to estimate the trend before policy introduction, and how this trend changed after policy introduction, also allowing for an immediate step change at the date the policy was introduced [18]. The post-intervention counterfactual was estimated as the continuation of the pre-policy introduction period trend. Absolute and relative differences were calculated at 3 and 5 years post-policy introduction in the control group and the intervention group counterfactual.

**Controlled interrupted time series.** Outcomes for intervention and control groups settings were further compared using segmented linear regression of the differences between the

groups [18,25]. The difference between the rate of knee replacement surgery in intervention and control CCG groups was calculated for each quarter and models were fitted to combined data from the pre- and post-intervention periods. The difference between the rate of surgery in the intervention group and its counterfactual value for each quarter in the period after policy introduction was calculated; the counterfactual was estimated as the continuation of the pre-policy introduction period trend.

**Pooled analysis.** Interrupted time series analysis was conducted for each CCG individually. Visual assessment of these graphs of quarterly rates during the study period showed no 'level change' in rates of operations evident after policy introduction. Instead, post-policy introduction trends for the change in slope in rates showed a strong effect for the majority of intervention CCGs. This was considered the 'effect size'. Random effects meta-analysis was used to pool the change in slope across CCG groups, stratifying according to whether the CCG policy was mild, moderate or strict. Data are presented as Forest Plots.

Data on rates of surgery for all intervention CCGs were then pooled, with the policy introduction date being considered time 'zero' for alignment in each CCG. A single segmented linear regression model was then fitted to obtain an overall national effect for all CCGs in England of the impact of health optimization policy introduction. The Newey-West standard error model was used to address the autocorrelation in the data detected with the Durbin-Watson test ($P < 0.001$) [26,27]. To control for secular effects, non-policy control CCGs were randomly matched to policy CCGs and assigned their policy start date. Policy and non-policy CCG data were then pooled, and a controlled interrupted time series analysis conducted, to compare difference in trends before and after policy introduction for an overall national effect of intervention compared to control CCGs.

Stratifications of the trends in surgery data for the time series analyses were conducted by BMI group, IMD category, and public versus privately funded operations.

All statistical analyses were conducted using Stata/MP version 16.1. The analyses were developed and reported according to the RECORD extension [28] to STROBE guidelines for observational studies using routinely collected data.

## Patient and public involvement

The Patient Experience Partnership in Research (PEP-R) group is a regional facilitated group [29], most of whom have had joint replacement, that provide patient and public input into research. Engagement with PEP-R in preparation for proposal of this research revealed surprise that 'this work hasn't been done already' and a feeling that it is 'vital to provide patients with evidence for the benefits of these policies'. Further engagement with the group during study design and analysis shaped the categorisation of policy severity. The group will also be engaged in planning the dissemination of the study results.

## Results

### Descriptive information and demographics

Of the 181 CCGs in continuous existence from 2013 to 2019, 46 (25.4%) were excluded due to incomplete policy information or complex policy activity timelines (e.g., stops and starts to policy use). 130 CCGs were included in the analyses, of which 74 (56.9%) had no policy (control CCGs), and 56 (43.1%) had a policy (intervention CCGs). Of those with policies: 26 (46.4%) had mild (advice only) policies, 14 (25.0%) had moderate (extra waiting time) policies and 16 (28.6%) had severe (mandatory BMI threshold) policies. Policy introduction dates ranged from mid-2013 to mid-2018. S1 Table details the CCGs included in the analysis, their policy types and start dates.

Within these CCGs, a total of 481,555 patients aged 40+ years had a primary total or uni-compartmental knee replacement between January 2009 and December 2019 in England, with osteoarthritis as a primary reason for surgery. The mean age of patients was 69.6 years (SD 9.13) and 275,626 (57.2%) were women. BMI was not recorded for 25.3% of patients. The mean BMI of patients with a BMI record was 30.9 kg/m² (SD 5.46), 431,856 (89.7%) operations were publicly funded, and 28,496 (5.9%) patients who received operations were from the 10% of most deprived areas.

Overall rates of surgery increased over time from 42.2 per 100,000 population aged 40+ per quarter year in 2009 to a peak of 75.7 in 2017, before declining to 59.6 in 2019. This was consistent across intervention and control CCG localities. There were approximately 11,000 operations in each quarter across control and intervention CCGs in total.

## Primary outcome in intervention CCGs: Patterns in rate of surgery following policy introduction

Interrupted time series analysis for individual CCGs in the intervention group (n = 56) showed heterogeneity in the effect of policy introduction on the rate of knee replacement operations. Where a change in trend was observed it was consistent with the time point of policy introduction identified *a priori*. *Fig 1* illustrates the heterogeneity in effect sizes on a caterpillar plot. Effect sizes ranged from a change in post-introduction from pre-introduction trend in rate of operations of -4.65 to +2.27. Most CCGs (75%) had a decrease in rate of operations following policy introduction (effect size estimate <0), and two CCGs (4%) showed evidence of an increase in rate of operations (effect size estimate 95% C.I lower bound >0).

In meta-analysis, the overall effect size of policy introduction was -0.92 (95% CI -0.57 to -1.29) operations per quarter per 100,000 patients aged 40+ years. Effect size was not

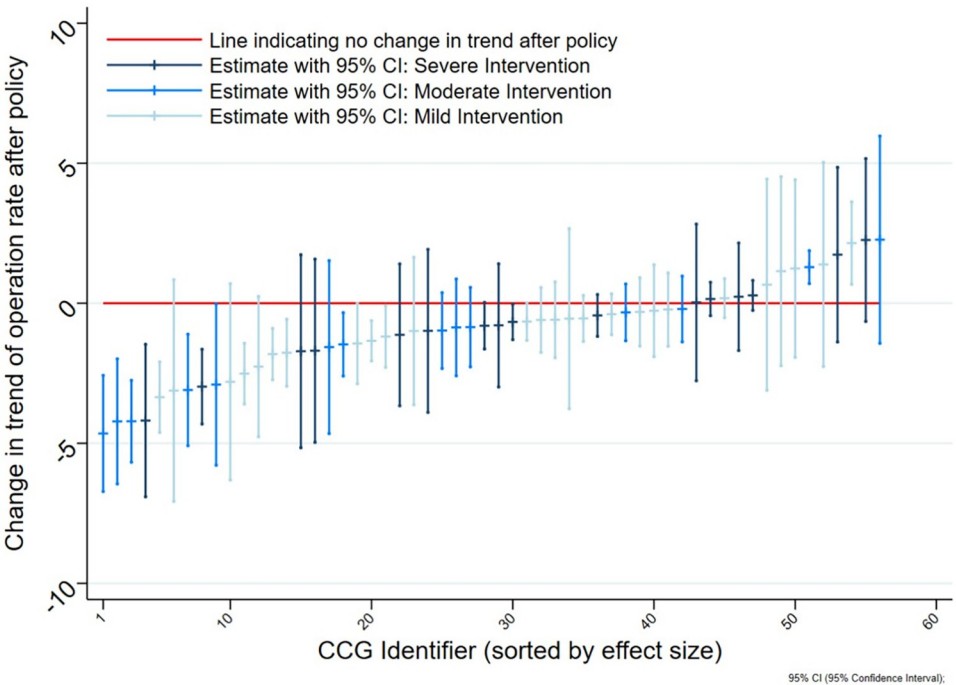

**Fig 1. Caterpillar plot of effect size in CCGs with policies of any severity n = 56 ('effect size' is regression model coefficient for change in pre- to post- policy introduction trends in rate of knee arthroplasty operations per 100,000 population aged 40+, per quarter).**

substantially associated with policy severity. *Fig 2* shows the effect sizes for CCGs in a meta-analysis and presented in groups of policy severity. Policy severity is categorised as: 1 (mild–patients receive advice only), 2 (moderate–patients are subject to additional waiting time before surgery) or 3 (strict–patients must be below a BMI threshold to be eligible for surgery).

## Comparison of outcome in control and intervention CCGs

The interrupted time series analysis of pooled data for all intervention and all control CCGs with alignment of their policy start dates is presented in *Fig 3*, illustrating the trend in operation rates pre- and post-policy introduction for the control and intervention CCGs. Before policy introduction both the intervention and control CCGs had an increasing trend in the rate of primary knee replacement surgery per 100,000 population aged 40+ per quarter. Intervention CCGs had a higher rate of surgery than the control CCGs in any given quarter before policy introduction. From the point of policy introduction, control group CCGs had no directional change in their trend; rate of surgery continued to increase over time, although at a reduced rate (*Table 1*). In contrast, for the intervention CCGs there was a reversal in trend at the point of policy introduction, which was sustained over time resulting in the mean rate of surgery becoming lower for intervention CCGs in any given quarter than for control CCGs. The immediate change in slope observed after policy introduction for each CCG was independent of differences in the date of policy introduction (e.g. the same effect was observed for a CCG introducing a policy in 2014, as for a CCG introducing the policy in 2018). There was no evidence that intervention CCGs had a level change in the rate of operations immediately following policy introduction.

*Table 1* presents the interrupted time series segmented linear regression model outputs for the control and intervention CCGs. There was strong evidence that there was a change in trend from the pre- to post-policy introduction period for the intervention CCGs: trend change -0.98 per quarter, 95% confidence interval (CI) -1.22 to -0.74, P<0.001.

For illustration, at 3 years after policy introduction, the modelled rate of operations per 100,000 aged 40+ per quarter in the intervention group was 64.1. This is a 4.6% reduction from the rate at the time of policy introduction (67.2). The predicted rate at 3 years in the counterfactual scenario (where the intervention group rate continued at the preintervention trend) is 74.6; an 11% increase from the rate at the time of policy introduction. The actual rate at 3 years in the intervention group was therefore 14.1% less than would have been expected had no policy introduction occurred. The modelled actual rate at 5 years is 59.9; 10.9% lower than the rate at the time of policy introduction, and 23.5% lower than the predicted counterfactual rate at 5 years (78.3).

The controlled interrupted time series difference-in-differences analyses results are also presented in *Table 1*. They indicate that the rate of knee replacement operations decreased by an additional 0.56 (95% confidence interval -0.76 to -0.36) operations per 100,000 aged 40 + per quarter in the intervention CCGs compared to the control CCGs. Compared to the counterfactual position the rate was decreased by an additional 1.00 (95% confidence interval 0.87 to 1.13) operation per 100,000 aged 40+ per quarter in the intervention CCGs.

## Baseline differences between intervention and control CCG groups

Intervention group CCGs had higher mean baseline rates (per 100,000 aged 40+) of surgery (2009 quarter 2), than those which did not; 47.3 (SD 16.2) compared to 38.2 (SD 16.1). *Table 2* shows the differences between the groups when 'baseline' is considered to be 18 months before the policy introduction date. CCGs that went on to introduce policies had patients who were more affluent, similarly obese, and more independently funded operations. The 'policy

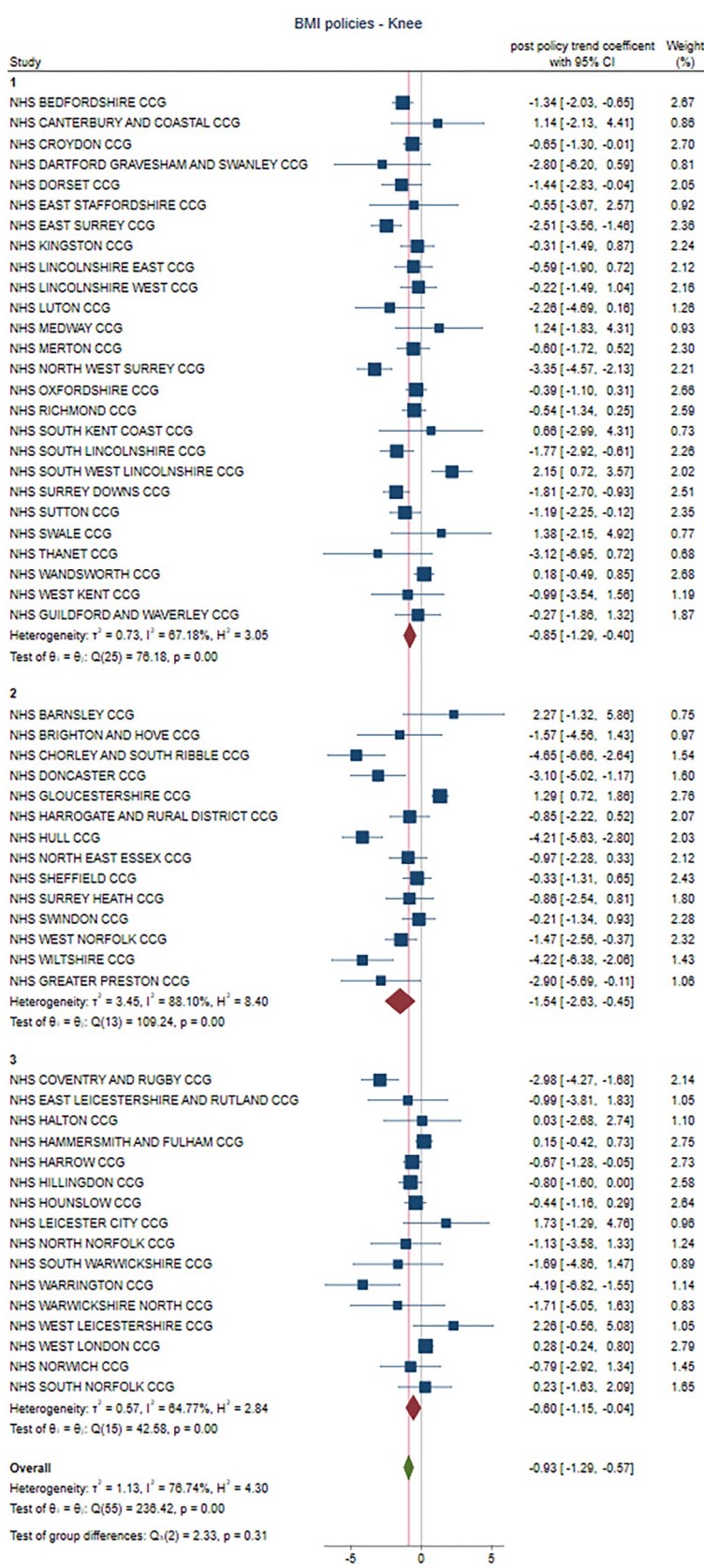

BMI policies - Knee

**Fig 2. Forest plot of effect size (as above) by policy category (1 = least severe) and with overall meta-analysis result for the intervention CCGs.**

introduction date' for control CCGs is the date of policy introduction from a randomly paired intervention CCG.

## Changes in patient characteristics after policy introduction

There were significant changes in patient characteristics after policy introduction in intervention CCGs, indicating that there was a differential impact of policies on patient groups. *Table 2* presents the patient characteristics in the CCGs at baseline, at 18-months post-policy introduction and at 3-years post-policy introduction. Post-policy introduction, patients in intervention

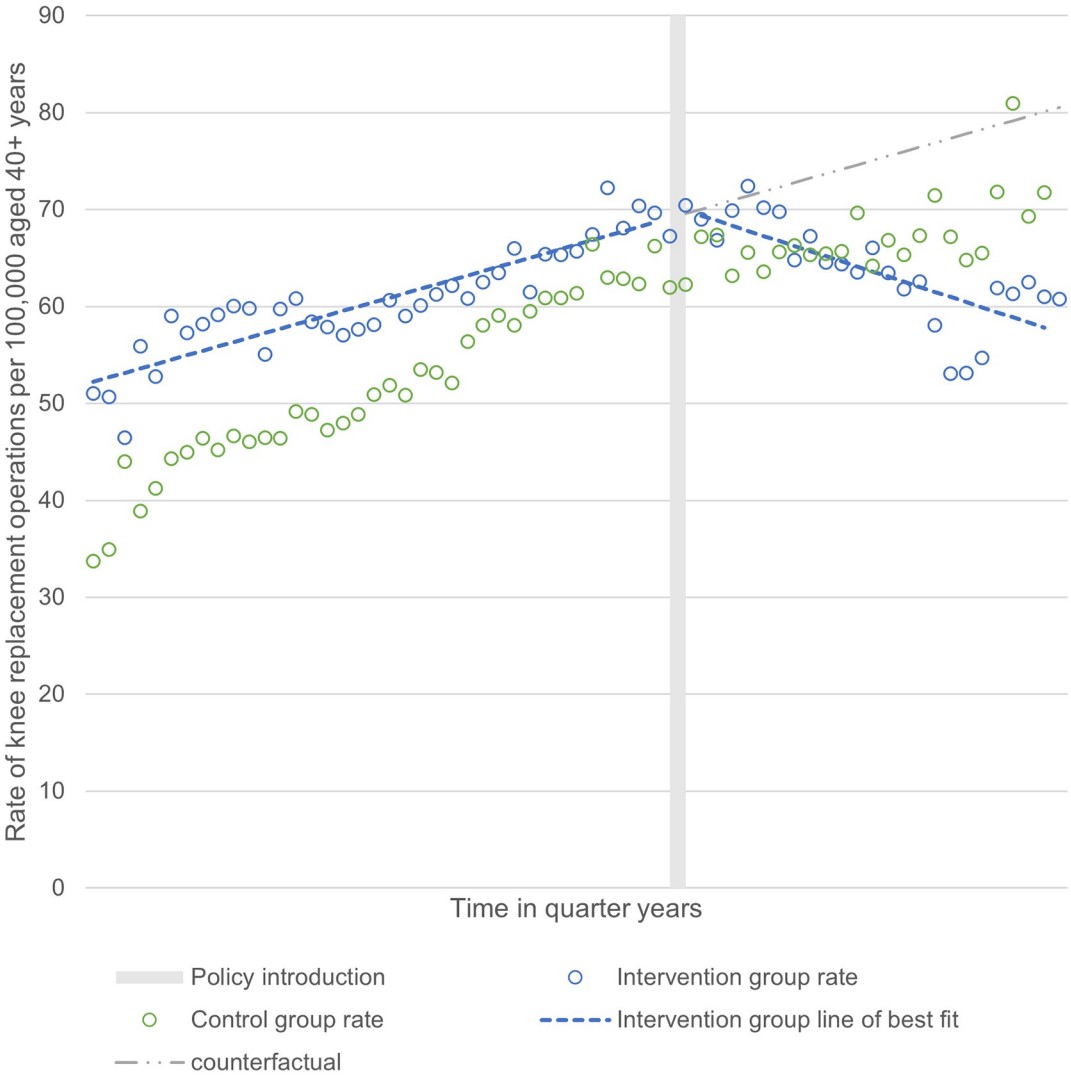

**Fig 3. Interrupted time series analyses of rate of knee replacement surgery per 100,000 population aged 40+ from pooled data for all intervention and control CCGs (n = 130).**

**Table 1. Interrupted time series segmented linear regression and difference in difference analyses before and after policy introduction in intervention and control CCGs.**

| Outcome | | Pre-policy introduction period | | | Policy introduction | | | Post-policy introduction period | | | | | |
|---|---|---|---|---|---|---|---|---|---|---|---|---|---|
| | | Quarterly trend | 95% CI | | Level change | 95% CI | | Quarterly trend | 95% CI | | Change in quarterly trend compared to pre-intervention | 95% CI | |
| **Rate of knee replacement surgery in 100,000 population aged 40+ years** | Intervention | 0.46 | 0.36 | 0.55 | 1.30 | -1.56 | 4.16 | -0.52 | -0.76 | -0.29 | -0.98 | -1.22 | -0.74 |
| | Control | 0.76 | 0.68 | 0.83 | -2.97 | -5.53 | -0.42 | 0.34 | 0.17 | 0.50 | -0.42 | -0.57 | -0.27 |
| | Difference in differences; intervention rate minus control rate | -0.30 | -0.40 | -0.20 | 4.28 | 0.89 | 7.66 | -0.86 | -1.07 | -0.65 | -0.56 | -0.76 | -0.36 |

CCGs were more likely to be: less deprived, higher American Society of Anesthesiologists (ASA) grade and privately funded.

Stratification of the interrupted time series analyses of pooled data for all intervention CCGs (displayed in *Fig* 4) showed the policy introduction was associated with a reduction in

**Table 2. Operation rate and patient characteristics of intervention and control CCGs pre- and post- policy introduction.**

| Operation and patient characteristics | Control CCGs (no policy introduced during study period) | | | Intervention CCGs (policy introduced during study period) | | |
|---|---|---|---|---|---|---|
| | baseline 18m pre | 18m post | 3y post | baseline 18m pre | 18m post | 3y post |
| | N = 74 | N = 74 | N = 37 | N = 56 | N = 56 | N = 30 |
| Knee replacement operations rate per 100,000 population aged 40+years per quarter (mean) | 61.36 | 63.58 | 69.65 | 65.69 | 70.19 | 63.55 |
| Age (mean) | 69.35 | 69.42 | 68.85 | 69.86 | 69.82 | 69.94 |
| Gender (% male) | 41.8% | 42.0% | 39.9% | 41.2% | 42.4% | 39.2% |
| BMI missing (%) | 27.9% | 20.8% | 22.2% | 23.9% | 21.5% | 22.9% |
| BMI (mean kg/m$^2$) | 31.23 | 30.82 | 31.05 | 31.12 | 30.76 | 30.76 |
| Underweight: BMI below 18 kg/m$^2$ (%) | 0.1% | 0.1% | 0.1% | 0.0% | 0.0% | 0.0% |
| Healthy weight: BMI 18 to 24.9 kg/m$^2$ (%) | 8.9% | 10.2% | 9.7% | 10.7% | 9.1% | 9.1% |
| Overweight; BMI 25 to 29.9 kg/m$^2$ (%) | 33.0% | 33.4% | 31.9% | 33.2% | 35.0% | 35.3% |
| Obese category 1: BMI 30 to 34.9 kg/m$^2$ (%) | 32.2% | 32.1% | 30.9% | 32.3% | 32.3% | 31.9% |
| Obese category 2: BMI 35 to 39.9 kg/m2 (%) | 17.7% | 17.9% | 18.4% | 16.1% | 17.7% | 15.6% |
| Obese category 3: BMI 40+ kg/m2 (%) | 8.2% | 6.3% | 9.1% | 7.7% | 5.9% | 8.1% |
| Independently funded surgery (%) | 8.9% | 10.3% | 8.3% | 11.1% | 12.5% | 13.8% |
| ASA* Grade (mean) | 2.10 | 2.14 | 2.14 | 2.08 | 2.14 | 2.11 |
| 1 –normal health (%) | 8.4% | 6.9% | 7.9% | 8.7% | 7.4% | 8.2% |
| 2 (%) | 73.7% | 72.3% | 70.8% | 74.7% | 71.1% | 72.6% |
| 3, 4 or 5 –poorest health (%) | 17.8% | 20.8% | 21.4% | 16.7% | 21.5% | 19.3% |
| Index of Multiple Deprivation (mean score) | 16026 | 16158 | 15787 | 18979 | 18919 | 19728 |
| Least deprived 20% | 17.5% | 17.4% | 17.9% | 25.6% | 24.7% | 29.5% |
| Less deprived 20–40% | 19.3% | 21.3% | 18.6% | 25.2% | 25.4% | 23.8% |
| Mid 20% deprived | 21.3% | 20.0% | 18.9% | 22.1% | 22.5% | 20.9% |
| More deprived 20–40% | 24.0% | 22.5% | 25.3% | 16.3% | 16.4% | 16.80 |
| Most deprived 20% | 17.8% | 18.8% | 19.3% | 10.8% | 11.1% | 8.9% |

* American Society of Anesthesiologists.

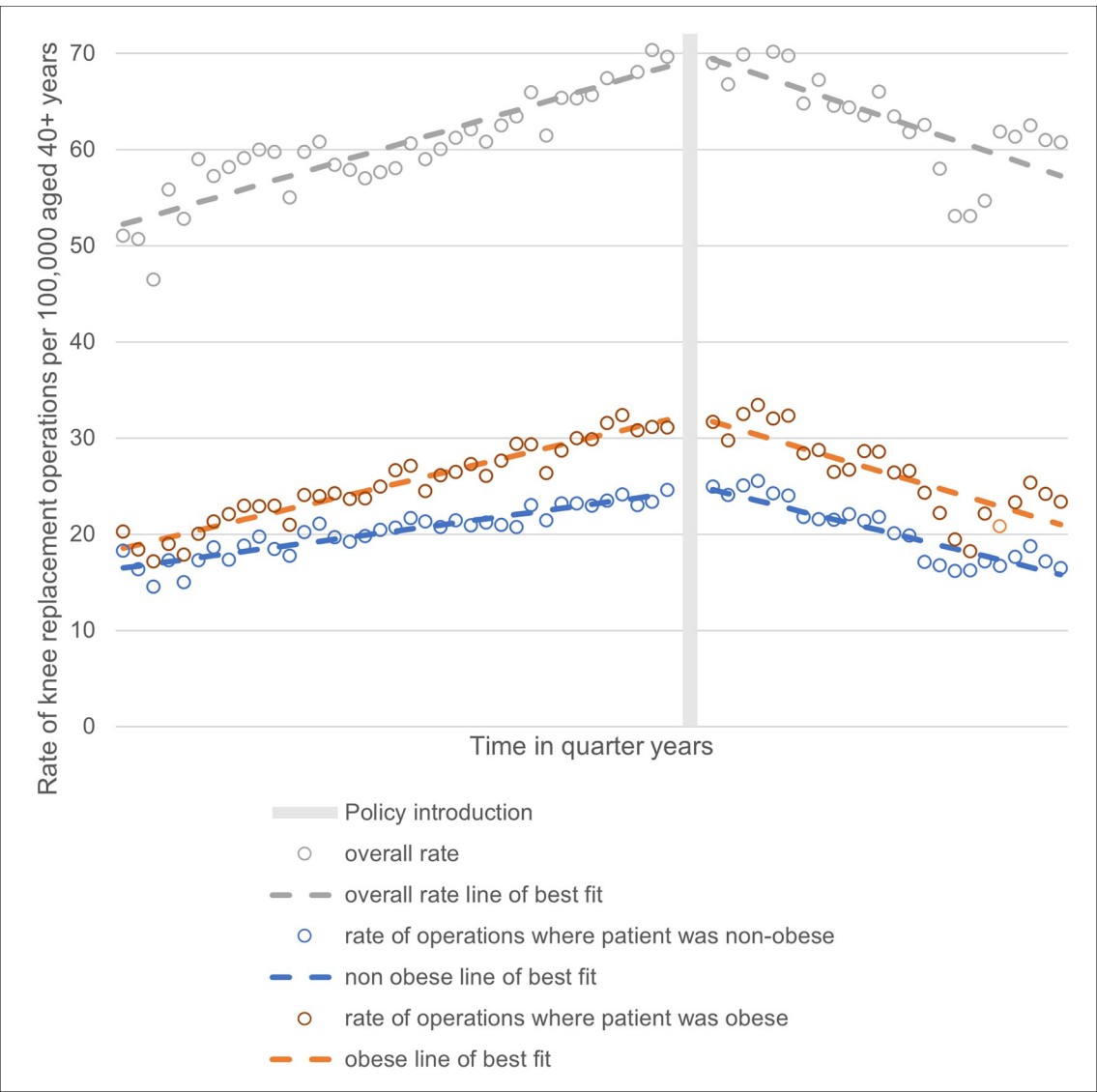

**Fig 4. Stratification of rate of knee replacement operations by obesity in intervention CCGs.**

the rate of operations done for all demographic groups, including in non-obese patients–a group which should not have been affected by the BMI policies. The denominator in each rate is the total CCG population aged 40+.

## Discussion

Introduction of a BMI health optimisation policy for knee replacement surgery is associated with a significant downward trend in rate of knee operations. This post-policy change amounts to approximately one fewer operation per 100,000 of the population aged 40+ years per quarter, representing a 14.1% reduction in the rate of surgery after 3 years compared to what would have been expected. There is no dose response effect evident with more severe policies. An unexpected decrease in operations for non-obese patients alongside obese patients was also observed. After policy introduction, patients receiving surgery are more likely to be less

socioeconomically deprived, independently funded and have a higher comorbidity score (ASA grade).

A reduction in the rate of surgery may represent a decrease in need for surgery, inappropriate restriction in access to surgery, or a combination of both. Health optimisation policies may reduce the need for surgery by supporting symptom improvement and quality of life through several mechanisms including weight loss, increased exercise and the opportunity for shared decision making [8]. Qualitative investigation into patients' experiences will be necessary to understand the mechanism of effect, however evidence shows ≥10% weight loss is needed for substantial symptom improvement in knee osteoarthritis [30] and so the reduction in rate of surgery seen here is unlikely to be accounted for through weight loss alone. The policies may have prevented access to surgery for patients in need of surgery, but who were unable or unwilling to lose sufficient weight to reach eligibility thresholds. Studies suggest that patients with BMI 40+ rarely find it possible to lose significant weight through lifestyle and pharmacological interventions alone when advised to do so for surgery, and that their response to being asked to lose weight may be to cease their pursuit of care for their joint symptoms despite needing surgery [31,32]. This may account for some of the reduction in rate of surgery in the obese patient group.

Despite National Institute for Health and Care Excellence (NICE) guidance stating that obesity should not preclude referral to surgery in osteoarthritis, it has been reported previously that CCG referral criteria are inconsistent in respect of NICE guidance [15]. There is no consistent evidence that patients with obesity have substantially worse outcomes from joint replacement surgery [33–35], nor that weight loss before joint replacement surgery has any effect on infection or readmission rates [36–38]. A study of arthroplasty patients in the USA suggests that using even a high BMI threshold of 40 kg/m$^2$ may prevent one operation with complications yet deny complication-free operations to 14 others [39]. While policies to limit access for obese patients may be driven by short term financial pressures, there is currently no evidence that treatment should be withheld on cost-effectiveness grounds. Economic modelling, which did not assess BMI, has concluded that compared to no arthroplasty, knee replacement arthroplasty was cost-effective for 99.9% of patients receiving surgery [40]. Given that at least approximately 45% of these patients are obese, and the overall context is of under-provision of surgery [41], the basis for local policies to ration this treatment appear limited. The need for surgery is higher in patients of lower socioeconomic status, and evidence that BMI eligibility criteria for joint replacement may worsen racial and socioeconomic disparities has been reported previously [42]. Data from this study show rates decreased most in more deprived groups (data on ethnicity were not available). There was also an association of policy introduction with an increase in the proportion of privately funded operations.

This study has used a powerful quasi-experimental design. Pooled data between 130 CCGs, with alignment of policy start dates which were spread over many different years, provides a robust reduction in bias from pre-intervention trends and secular trends. A further strength of this study is the use of a large mandatory national dataset, capturing 96% of all knee replacement procedures including those that are independently funded [5], and for this study the IMD 2015 was linked to all patients. BMI data are less complete in the registry–missing for approximately 25% of records. Some surgery eligibility policies included restrictions on patients who smoke. As the NJR does not collect data on smoking status, no analysis was possible on this. The COVID-19 pandemic's impact on elective surgery has been significant and alters the immediate applicability of the study findings, however the imminent formation of Integrated Care Systems from CCGs represents a key opportunity for policy makers to be aware of the potential to widen health inequalities. Our finding of no dose-response relationship between severity of the policy and change in rate of surgery, suggests that even a modestly

restrictive policy may exacerbate inequality of access to knee replacement. As our primary data source is a registry of surgery, this study cannot comment on the patients who did not receive surgery through choice or exclusion. Research is needed that determines the impact of policies on this group as they are at high risk of health inequality. Examination of policy implementation rigour and resource may elucidate the reasons behind the heterogeneity in effect size seen in this study, and the unexpected impact on non-obese patients. With the concerns over the unintended effects of health optimisation policies that target access to surgery raised here, policies could instead focus on supporting long-term lifestyle changes within existing waiting times, avoiding the risks of punitive restrictions on access to surgery [9,43].

## Conclusion

In summary our study has reported strong evidence that commissioning policies for body mass index that alter access to surgery for knee arthroplasty are followed by a reduction in the rate of surgery, though the mechanism for this reduction is not yet understood. Stratification of data in this study suggests that the policies may be worsening health inequalities by reducing the number of operations provided to socioeconomically deprived patients as well as driving patients towards independently funded surgery. There is also evidence of unintended effects such as a reduction in the rate of surgery for non-obese patients which requires further investigation.

## Supporting information

**S1 Table. Details of clinical commissioning group policies on weight loss and body mass index thresholds for knee replacement surgery for CCGs in existence from Jan 2013 to Dec 2019.** Policies started less than 18 months prior to Dec 2019 are not included.
(PDF)

## Acknowledgments

We thank the patients and staff of all the hospitals in England, Wales and Northern Ireland who have contributed data to the National Joint Registry (NJR). We are grateful to the Healthcare Quality Improvement Partnership (HQIP), the NJR Research Committee and staff at the NJR Centre for facilitating this work. The authors have conformed to the NJR's standard protocol for data access and publication. The views expressed represent those of the authors and do not necessarily reflect those of the National Joint Registry Steering Committee or the Healthcare Quality Improvement Partnership (HQIP) who do not vouch for how the information is presented.

HQIP and the NJR take no responsibility for the accuracy, currency, reliability and correctness of any data used or referred to in this report, nor for the accuracy, currency, reliability and correctness of links or references to other information sources and disclaims all warranties in relation to such data, links and references to the maximum extent permitted by legislation. HQIP and NJR shall have no liability (including but not limited to liability by reason of negligence) for any loss, damage, cost or expense incurred or arising by reason of any person using or relying on the data within this report and whether caused by reason of any error, omission or misrepresentation in the report or otherwise. This report is not to be taken as advice. Third parties using or relying on the data in this report do so at their own risk and will be responsible for making their own assessment and should verify all relevant representations, statements and information with their own professional advisers.

## Contributor and guarantor information

All authors contributed to the study design. AJ and SH provided expert statistical input. All authors had full access to all statistical reports and tables in the study. JM had full access to all of the study data and takes responsibility for the integrity of the data and the accuracy of the data analysis. JM wrote the first draft. All authors contributed to the interpretation of results and critical revision of the manuscript and approved the final manuscript. The corresponding author attests that all listed authors meet authorship criteria and that no others meeting the criteria have been omitted.

## Transparency declaration

AJ is the manuscript's guarantor, and affirms that this manuscript is an honest, accurate, and transparent account of the study being reported; that no important aspects of the study have been omitted; and that any discrepancies from the study as planned (and, if relevant, registered) have been explained.

## Copyright/Licence for publication

The Corresponding Author has the right to grant on behalf of all authors and does grant on behalf of all authors, a worldwide licence to the Publishers and its licensees in perpetuity, in all forms, formats and media (whether known now or created in the future), to i) publish, reproduce, distribute, display and store the Contribution, ii) translate the Contribution into other languages, create adaptations, reprints, include within collections and create summaries, extracts and/or, abstracts of the Contribution, iii) create any other derivative work(s) based on the Contribution, iv) to exploit all subsidiary rights in the Contribution, v) the inclusion of electronic links from the Contribution to third party material where-ever it may be located; and, vi) licence any third party to do any or all of the above.

## Author Contributions

**Conceptualization:** Joanna McLaughlin, J Mark Wilkinson, Andrew Judge.

**Data curation:** Joanna McLaughlin.

**Formal analysis:** Joanna McLaughlin, Andrew Judge.

**Funding acquisition:** Joanna McLaughlin, Andrew Judge.

**Investigation:** Joanna McLaughlin.

**Methodology:** Samuel Hawley, J Mark Wilkinson, Andrew Judge.

**Project administration:** Joanna McLaughlin.

**Supervision:** Ruth Kipping, Amanda Owen-Smith, Hugh McLeod, Andrew Judge.

**Writing – original draft:** Joanna McLaughlin, Andrew Judge.

**Writing – review & editing:** Joanna McLaughlin, Ruth Kipping, Amanda Owen-Smith, Hugh McLeod, Samuel Hawley, J Mark Wilkinson, Andrew Judge.

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
