## [Decision Letter · Decision Letter 0]

8 Jun 2022

What effect have NHS commissioners’ policies for body mass index had on access to knee replacement surgery in England?: An interrupted time series analysis from the National Joint Registry

PONE-D-22-12380

Dear Dr. McLaughlin

We’re pleased to inform you that your manuscript has been judged scientifically suitable for publication and will be formally accepted for publication once it meets all outstanding technical requirements.

Kind regards,

Louisa Ells, Ph.D.

Academic Editor

PLOS ONE

1. Thank you for stating the following in the Competing Interests section: 

[I have read the journal's policy and the authors of this manuscript have the following competing interests:

All authors have completed the Unified Competing Interest form at www.icmje.org/coi_disclosure.pdf and declare: AJ has received consultancy fees from Freshfields Bruckhaus Deringer and has held advisory board positions (which involved receipt of fees) from Anthera Pharmaceuticals, INC, outside the submitted work; no other relationships or activities that could appear to have influenced the submitted work.]

Please respond by return email with your amended Competing Interests Statement and we will change the online submission form on your behalf.

3. We note that you have included the phrase “data not presented” in your manuscript. Unfortunately, this does not meet our data sharing requirements. PLOS does not permit references to inaccessible data. We require that authors provide all relevant data within the paper, Supporting Information files, or in an acceptable, public repository. Please add a citation to support this phrase or upload the data that corresponds with these findings to a stable repository (such as Figshare or Dryad) and provide and URLs, DOIs, or accession numbers that may be used to access these data. Or, if the data are not a core part of the research being presented in your study, we ask that you remove the phrase that refers to these data.

Additional Editor Comments:

A well written and important manuscript.

Reviewers' comments:

Reviewer's Responses to Questions

**Comments to the Author**

1. Is the manuscript technically sound, and do the data support the conclusions?

Reviewer #1: Yes

Reviewer #2: Yes

2. Has the statistical analysis been performed appropriately and rigorously? 

Reviewer #1: I Don't Know

Reviewer #2: Yes

3. Have the authors made all data underlying the findings in their manuscript fully available?

Reviewer #1: No

Reviewer #2: Yes

4. Is the manuscript presented in an intelligible fashion and written in standard English?

Reviewer #1: Yes

Reviewer #2: Yes

5. Review Comments to the Author

Reviewer #1: This manuscript explores the effect that NHS commissioners’ policies for body mass index had on access to knee replacement surgery in England. It is a very well-written manuscript, with an important and striking message. The results are clearly presented and well discussed. I have no major or minor comments to improve the manuscript.

Reviewer #2: This is a very interesting and rigorously conducted piece of work and the authors are to be commended. It is important that we understand the effects of policy on the provision of services to patients.

6. PLOS authors have the option to publish the peer review history of their article (what does this mean?). If published, this will include your full peer review and any attached files.

Reviewer #1: No

Reviewer #2: No

---

## [Editor Report · Acceptance letter]

14 Jun 2022

PONE-D-22-12380 

What effect have NHS commissioners’ policies for body mass index had on access to knee replacement surgery in England?: An interrupted time series analysis from the National Joint Registry 

Dear Dr. McLaughlin:

I'm pleased to inform you that your manuscript has been deemed suitable for publication in PLOS ONE. Congratulations! Your manuscript is now with our production department. 

Kind regards, 

on behalf of

Prof Louisa Ells 

Academic Editor

PLOS ONE